# An Efficient Pessimistic-Optimistic Algorithm for Stochastic Linear Bandits with General Constraints

**Xin Liu**
University of Michigan, Ann Arbor
xinliuee@umich.edu

**Bin Li**
Pennsylvania State University
binli@psu.edu

**Pengyi Shi**
Purdue University
shi178@purdue.edu

**Lei Ying**
University of Michigan, Ann Arbor
leiying@umich.edu

## Abstract

This paper considers stochastic linear bandits with general nonlinear constraints. The objective is to maximize the expected cumulative reward over horizon $T$ subject to a set of constraints in each round $\tau \leqslant T$. We propose a pessimistic-optimistic algorithm for this problem, which is efficient in two aspects. First, the algorithm yields $\tilde{\mathcal{O}}\left(\left(\frac{K^{0.75}}{\delta} + d\right)\sqrt{\tau}\right)$ (pseudo) regret in round $\tau \leqslant T$, where $K$ is the number of constraints, $d$ is the dimension of the reward feature space, and $\delta$ is a Slater's constant; and *zero* constraint violation in any round $\tau > \tau'$, where $\tau'$ is *independent* of horizon $T$. Second, the algorithm is computationally efficient. Our algorithm is based on the primal-dual approach in optimization and includes two components. The primal component is similar to unconstrained stochastic linear bandits (our algorithm uses the linear upper confidence bound algorithm (LinUCB)). The computational complexity of the dual component depends on the number of constraints, but is independent of the sizes of the contextual space, the action space, and the feature space. Thus, the computational complexity of our algorithm is similar to LinUCB for unconstrained stochastic linear bandits.

## 1   Introduction

Stochastic linear bandits have a broad range of applications in practice, including online recommendations, job assignments in crowdsourcing, and clinical trials in healthcare. Most existing studies on stochastic linear bandits formulated them as unconstrained online optimization problems, limiting their application to problems with operational constraints such as safety, fairness, and budget constraints. In this paper, we consider a stochastic linear bandit with general constraints. As in a standard stochastic linear bandit, at the beginning of each round $t \in [T]$, the learner is given a context $c(t)$ that is randomly sampled from the context set $\mathcal{C}$ (a countable set), and takes an action $A(t) \in [J]$. The learner then receives a reward $R(c(t), A(t)) = r(c(t), A(t)) + \eta(t)$, where $r(c, j) = \langle \theta_*, \phi(c, j) \rangle$, $\phi(c, j) \in \mathbb{R}^d$ is a $d$-dimensional feature vector for (context, action) pair $(c, j)$, $\theta_* \in \mathbb{R}^d$ is an unknown underlying vector to be learned, and $\eta(t)$ is a zero-mean random variable. For constrained stochastic linear bandits, we further assume when action $A(t)$ is taken on context $c(t)$, it incurs $K$ different types of costs, denoted by $W^{(k)}(c(t), A(t))$. We assume $W^{(k)}(c, j)$ is a random variable with mean $w^{(k)}(c, j)$ that is unknown to the learner. This paper considers general cost functions and does *not* require $w^{(k)}(c, j)$ to have a linear form like $r(c, j)$.

Denote the action taken by policy $\pi$ in round $t$ by $A^\pi(t)$. The learner's objective is to learn a policy $\pi$ that maximizes the cumulative rewards over horizon $T$ subject to *anytime cumulative constraints*:

$$\max_{\pi} \mathbb{E}\left[\sum_{t=1}^{T} R(c(t), A^{\pi}(t))\right] \quad (1) \qquad \mathbb{E}\left[\sum_{t=1}^{\tau} W^{(k)}(c(t), A^{\pi}(t))\right] \leqslant 0, \forall \tau \in [T], k \in [K]. \quad (2)$$

The constraint (2) above may represent different operational constraints including safety, fairness, and budget constraints.

**Anytime cumulative constraints**

In the literature, constraints in stochastic bandits have been formulated differently. There are two popular formulations. The first one is a cumulative constraint over horizon $T$, including knapsack bandits [10, 9, 3, 4, 5, 17, 11] where the process terminates when the total budget has been consumed; fair bandits where the number of times an action can be taken must exceed a threshold at the end of the horizon [12]; and contextual bandits with a cumulative budget constraint [40, 14]. In these settings, the feasible action set in each round depends on the history. In general, the learner has more flexibility in the earlier rounds, close to that in the unconstrained setting. Another formulation is *anytime* constraints, which either require the expected cost of the action taken in each round to be lower than a threshold [6, 29] or the expected cost of the policy in each round is lower than a threshold [32]. We call them *anytime* action constraints and *anytime* policy constraints, respectively.

Our constraint in the form of (2) is an *anytime cumulative constraint*, i.e., it imposes a cumulative constraint in *every* round. This anytime cumulative constraint is most similar to the anytime policy constraint in [32] because the average cost of a policy is close to its mean after the policy has been applied for many rounds and the process converges, so it can be viewed as a cumulative constraint on actions over many rounds (like ours). Furthermore, when our anytime cumulative constraint (2) is satisfied, it is guaranteed that the time-average cost is below a threshold in every round.

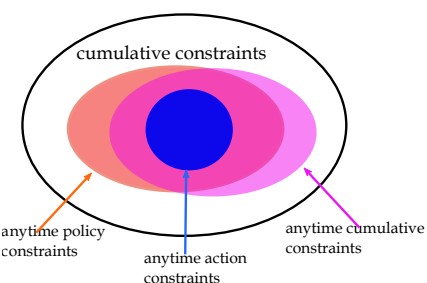

Figure 1: A conceptual description of feasible policy sets under different constraint formulations.

In summary, our anytime cumulative constraint is stricter than a cumulative constraint over fixed horizon $T$ but is less restrictive than anytime action constraint in [6, 29]. Figure 1 provides a conceptual description of the relationship between these different forms of constraints.

**Main Contributions**

This paper presents a pessimistic-optimistic algorithm based on the primal-dual approach in optimization for the problem defined in (1)-(2). The algorithm is efficient in two aspects. First, the algorithm yields $\tilde{\mathcal{O}}\left(\left(\frac{K^{0.75}}{\delta} + d\right)\sqrt{\tau}\right)$ regret in round $\tau \leqslant T$ and achieves *zero* constraint violation in any round $\tau > \tau'$ for a constant $\tau'$ independent of horizon $T$. Second, the algorithm is computationally efficient.

For computational efficiency, the design of our algorithm is based on the primal-dual approach in optimization. The computation of the primary component is similar to unconstrained stochastic linear bandits [15, 33, 25, 1, 13]. The dual component includes a set of Lagrangian multipliers that are updated in a simple manner to keep track of the levels of constraint violations so far in each round; the update depends on the number of constraints, but it is independent of the sizes of the contextual space, the action space, and the feature space. Thus, the overall computational complexity of our algorithm is similar to that of LinUCB in the unconstrained setting. This results in a much more efficient calculation comparing to OPLB proposed in [32]. OPLB needs to construct a safe policy set in each round, hence, its computational complexity is prohibitively high as the authors acknowledged.

For constraint violation, our algorithm guarantees that for any $\tau > \tau'$, the constraint holds with probability one. In other words, after a constant number of rounds, the constraint is always satisfied. This is in contrast to prior works [32, 6], where anytime constraints are proven to hold over horizon $T$ with probability $1 - \chi$ for a constant $\chi$. In other words, the anytime constraints may be violated with probability $\chi$, and it is not clear how often they are violated when it happens. Furthermore,

beyond mean cost constraints considered in (2) and in [32, 6], we prove that a sample-path version of constraint (2) holds with probability $1 - O\left(e^{-\frac{\delta\sqrt{\tau}}{50K^{2.5}}}\right)$ in round $\tau$ under our algorithm.

To summarize, our algorithm is computationally efficient and provides strong guarantees on both regret and constraint violations. Additionally, our cost function is in a general form and does not need to be linear as those in [32, 6]. We discuss more related work in the following.

### Related Work

Stochastic linear bandits [2, 8] are a special class of contextual bandits [39, 23], which generalize multi-armed bandits [22]. Besides [32], [12] considered an adversarial contextual bandit with anytime policy constraint representing fairness. The proposed algorithm has $\tilde{\mathcal{O}}(\sqrt{|\mathcal{C}|JT})$ regret when the context distribution is known to the learner; otherwise it has $\tilde{\mathcal{O}}(\sqrt{|\mathcal{C}|JT})$ regret and $\tilde{\mathcal{O}}(\sqrt{|\mathcal{C}|T})$ constraint violation. [24] studied a combinatorial sleeping bandits problem under cumulative fairness constraints and proposed an algorithm based on UCB which they *conjectured* to have $\tilde{\mathcal{O}}(\sqrt{T})$ regret and $\tilde{\mathcal{O}}(\sqrt{T})$ constraint violation. Recent work studied unconstrained structured bandits and proposed primal-dual approach based on asymptotically lower bound problem in bandits [21, 35, 16]. However, our algorithm is different from them in three aspects. Our primal component is a greedy algorithm instead of a (sub-)gradient algorithm (as in [35]). Our dual component does not solve a best response problem, which is a constrained optimization problem as in [21, 16]. Our analysis is based on the Lyapunov-drift analysis for queueing systems, e.g., we establish a bound on the exponential moment of the dual variable, which is not present in [21, 35, 16]. It is also worth mentioning that [20, 19, 30] studied "conservative" bandits which require that the reward or the cumulative reward exceeds a threshold at each step. Another line of related work is online convex optimization with constraints, studied in [28, 27, 41, 42, 38, 36], where online primal-dual with proximal regularized algorithms have been proposed to achieve $O(\sqrt{T})$ regret and $O(1)$ violation for static constraints in [41] and $O(\sqrt{T})$ violation for stochastic constraints in [42] .

**Notation.** $f(n) = \tilde{\mathcal{O}}(g(n))$ denotes $f(n) = O(g(n)\log^k n)$ with $k > 0$; $[N]$ denotes the set $\{1, 2, \cdots, N\}$; $\langle \cdot, \cdot \rangle$ denotes the inner product; $(\cdot)^\dagger$ denotes the transpose of a vector or a matrix; $||\cdot|| = ||\cdot||_2$, and $||\mathbf{x}||_\Sigma = \sqrt{\mathbf{x}^\dagger \Sigma \mathbf{x}}$.

## 2  A Pessimistic-Optimistic Algorithm

We consider a stochastic linear bandit over horizon $T$ as described in the introduction. The learner's objective is to maximize the cumulative reward over time horizon $T$ subject to $K$ anytime cumulative constraints as defined in (1)-(2). To address the challenges on the *unknown* reward and cost in constraint, as well as the anytime cumulative constraints, we develop a pessimistic-optimistic algorithm based on the primal-dual approach for constrained optimization. We first give out the intuition of the algorithm and then provide the formal statement of the algorithm.

To start, we consider a baseline, deterministic problem that replaces all the random variables with their expectations. Different from the conventional setup, we introduce a "tightness" constant $\epsilon > 0$:

$$\max_{\mathbf{x}} \sum_{c \in \mathcal{C}, j \in [J]} p_c r(c, j) x_{c,j} \tag{3}$$

$$\text{s.t.} \sum_{j \in [J]} x_{c,j} = 1, \ x_{c,j} \geqslant 0, \forall c \in \mathcal{C}, \tag{4}$$

$$\sum_{c \in \mathcal{C}, j \in [J]} p_c w^{(k)}(c, j) x_{c,j} + \epsilon \leqslant 0, \ \forall k \in [K], \tag{5}$$

where $x_{c,j}$ can be viewed as the probability of taking action $j$ on context $c$, and $p_c$ is the probability that context $c$ is selected in each round. We will discuss in further details the importance of the tightness constant $\epsilon$ in Section 4. The Lagrangian of the problem above is

$$\max_{\mathbf{x}: \sum_j x_{c,j}=1, x_{c,j} \geqslant 0} \sum_{c,j} p_c r(c, j) x_{c,j} - \sum_k \lambda^{(k)} \left( \sum_{c \in \mathcal{C}, j \in [J]} p_c w^{(k)}(c, j) x_{c,j} + \epsilon \right), \tag{6}$$

where $\lambda^{(k)}$ is the Lagrange multiplier associated with the $k$th constraint in (5). Fixing the values of the Lagrange multipliers, solving the optimization problem is equivalent to solving $|\mathcal{C}|$ separate subproblems (7), one for each context $c$, because the optimization variables $\mathbf{x}$ are coupled through $j$ only:

$$\max_{\mathbf{x}:\sum_j x_{c,j}=1, x_{c,j} \geqslant 0} p_c \left( \sum_j r(c,j) x_{c,j} - \sum_k \lambda^{(k)} \left( \sum_j w^{(k)}(c,j) x_{c,j} \right) \right). \tag{7}$$

Since the problem above is a linear programming, one of the optimal solutions is $x_{c,j} = 1$ for $j = j^*$ and $x_{c,j} = 0$ otherwise, where

$$j^* \in \arg\max_j r(c,j) - \sum_k \lambda^{(k)} w^{(k)}(c,j) \tag{8}$$

and a tie can be broken arbitrarily. If we call $r(c,j) - \sum_k \lambda^{(k)} w^{(k)}(c,j)$ the action-value of context $c$, then the solution for fixed values of Lagrange multipliers is to choose an action with the highest action-value. We may view the action value here plays a similar role as the Q-value (also called action-value function) in Q-learning [37].

Now the challenges to find a solution according to (8) include: (i) both $r(c,j)$ and $w^{(k)}(c,j)$ are unknown, and (ii) the optimal Lagrange multipliers $\lambda^{(k)}$ are also unknown. To overcome these challenges, we develop a pessimistic-optimistic algorithm that

- Uses LinUCB to estimate $r(c,j)$ based on its linear structure.
- Uses observed $W^{(k)}(c(t),j)$ to replace $w^{(k)}(c(t),j)$ at each round $t$.
- Uses the following function to dynamically approximate the Lagrange multipliers ($X_j(t) = 1$ if $A(t) = j$; otherwise $X_j(t) = 0$):

$$Q^{(k)}(t+1) = \left[ Q^{(k)}(t) + \sum_{j\in[J]} W^{(k)}(c(t),j) X_j(t) + \epsilon_t \right]^+, \forall k.$$

In other words, we increase its value when the current cost exceeds the current "budget," and decrease it otherwise. Therefore, $Q^{(k)}(t)$ keeps track of the cumulative constraint violation by round $t$.

- We further add a scaling parameter $1/V_t$ to $Q^{(k)}(t)$, i.e. $Q^{(k)}(t)/V_t$, to approximate $\lambda^{(k)}$. With a carefully designed $V_t$, we can control the tradeoff between maximizing reward and minimizing constraint violation in the policy and achieve the regret and constraint violation bounds to be presented in the main theorem.

Next, we formally state our algorithm. This algorithm takes the following information as input at the beginning of each round $t$: (i) historical observations

$$\mathcal{F}_{t-1} = \{c(s), A(s), R(c(s), A(s)), W^{(k)}(c(s), A(s))\}_{s\in[t-1], k\in[K]},$$

(ii) current observations $c(s)$ and $\{W^{(k)}(c(s),j)\}_{k\in[K], j\in[J]}$, and (iii) system parameters: the feature map $\{\phi(c,j)\}_{c\in\mathcal{C}, j\in[J]}$, time horizon $T$, and a pre-set constant $\delta$. In the analysis of our algorithm, we will reveal the connection of this constant $\delta$ with Slater's condition. The algorithm outputs the action in each round, observes reward $R(c(t), A(t))$, makes updates, and then moves to the next round $t+1$.

**A Pessimistic-Optimistic Algorithm**

---

**Initialization:** $Q^{(k)}(1) = 0$, $\mathcal{B}_1 = \{\theta | ||\theta||_{\Sigma_0} \leqslant \sqrt{\beta_1}\}$, $\Sigma_0 = \mathbf{I}$ and $\sqrt{\beta_1} = m + \sqrt{2\log T}$.

For $t = 1, \cdots, T$,

- **Set:** $V_t = \delta K^{0.25}\sqrt{\frac{2t}{3}}$ and $\epsilon_t = K^{0.75}\sqrt{\frac{6}{t}}$.
- **LinUCB (Optimistic):** Use LinUCB to estimate $r(c(t), j)$ for all $j$:

$$\hat{r}(c(t), j) = \min\{1, \tilde{r}(c(t), j)\} \quad \text{with} \quad \tilde{r}(c(t), j) = \max_{\theta\in\mathcal{B}_t}\langle\theta, \phi(c(t), j)\rangle.$$

- **MaxValue:** Compute *pseudo-action-value* of context $c(t)$ for all action $j$, and take the action $A(t) = j^*$ with the highest pseudo-action-value, breaking a tie arbitrarily

$$j^* \in \arg\max_j \underbrace{\hat{r}(c(t), j) - \frac{1}{V_t} \sum_k W^{(k)}(c(t), j) Q^{(k)}(t)}_{\text{pseudo action value of } (c(t), j)}.$$

- **Dual Update (Pessimistic):** Update the estimates of dual variables $Q^{(k)}(t)$ as follows:

$$Q^{(k)}(t+1) = \left[ Q^{(k)}(t) + \sum_j W^{(k)}(c(t), j) X_j(t) + \epsilon_t \right]^+, \forall k. \tag{9}$$

- **Confidence Set Update:** Update $\Sigma_t$, $\hat{\theta}_t$, $\beta_{t+1}$ and $\mathcal{B}_{t+1}$ by the received reward $R(c(t), j^*)$ :

$$\Sigma_t = \Sigma_{t-1} + \phi(c(t), j^*) \phi^\dagger(c(t), j^*), \quad \hat{\theta}_t = \Sigma_t^{-1} \sum_{s=1}^t \phi(c(s), A(s)) R(c(s), A(s)),$$

$$\sqrt{\beta_{t+1}} = m + \sqrt{2 \log T + d \log \left( \frac{d+t}{d} \right)}, \quad \mathcal{B}_{t+1} = \{ \theta \mid ||\theta - \hat{\theta}_t||_{\Sigma_t} \leqslant \sqrt{\beta_{t+1}} \}.$$

---

The complexity of our algorithm is similar to LinUCB. The additional complexity is proportional to the number of constraints (for updating $Q^{(k)}$), and it is much lower than OPLB in [32], where the construction of a safe policy set in each round is a major computational hurdle. Our algorithm is computationally efficient. Additionally, our algorithm does not estimate $w^{(k)}(c, j)$, hence, we do not need to make any specific assumption on $w^{(k)}(c, j)$.

## 3  Main Results: Regret and Constraint Violation Bounds

To understand the performance of a given policy $\pi$, we will analyze both the regret and the constraint violation. For that, we first define the baselines and state the assumptions made for the performance analysis. Then, we present our main results on the regret bound and constraint violations – for the latter, we present both results on expected violation and an additional high probability bound for pathwise constraint violation.

### 3.1  Baselines and Assumptions

**Regret baseline:** We consider the following optimization problem:

$$\max_{\mathbf{x}} \sum_{c \in \mathcal{C}, j \in [J]} p_c r(c, j) x_{c,j} \tag{10}$$

$$\text{s.t.} \sum_{j \in [J]} x_{c,j} = 1, \ x_{c,j} \geqslant 0, \forall c \in \mathcal{C}, \tag{11}$$

$$\sum_{c \in \mathcal{C}, j \in [J]} p_c w^{(k)}(c, j) x_{c,j} \leqslant 0, \ \forall k \in [K]. \tag{12}$$

**Constraint violation baseline:** We choose zero (no violation) as our baseline.

It worth noting the baseline we use in the regret analysis is derived from relaxed cumulative constraints instead of anytime cumulative constraints in the original problem. Since the cumulative constraint is the least restrictive constraint, a learner obtains the highest cumulative rewards in such a setting. In other words, our regret analysis is with respect to the best (the most relaxed) baseline.

We make the following assumptions for all the results present in this paper.

**Assumption 1** *The context $c(t)$ are i.i.d. across rounds. The mean reward $r(c, j) = \langle \theta_*, \phi(c, j) \rangle \in [0, 1]$ with $||\phi(c, j)|| \leqslant 1, ||\theta_*|| \leqslant m$ for any $c \in \mathcal{C}, \ j \in [J]$. $\eta(t)$ is zero-mean 1-subGaussian conditioned on $\{\mathcal{F}_{t-1}, A(t)\}$.*

**Assumption 2** *The costs in the constraints satisfy $|W^{(k)}(c,j)| \leq 1$. Furthermore, we assume $\{W^{(k)}(c,j)\}_{t=1}^{T}$ are i.i.d. samples for given $c$ and $j$.*

**Assumption 3 (Slater's condition)** *There exists $\delta > 0$ such that there exists a feasible solution $\mathbf{x}$ to optimization problem (10)-(12) that guarantees $\sum_{c\in\mathcal{C},j\in[J]} p_c w^{(k)}(c,j)x_{c,j} \leq -\delta, \forall k \in [K]$. We assume $\delta \leq 1$ because if the condition holds for $\delta > 1$, it also holds for $\delta = 1$.*

We call $\delta$ Slater's constant because it comes from Slater's condition in optimization – this is the constant used as an input of our algorithm. This constant plays a similar role as the cost of a safe action in [6, 32]. In fact, a safe action guarantees the existence of a Slater's constant, and we can estimate the constant by running the safe action for a period of time. However, the existence of a Slater's constant does not require the existence of a safe action. It is also a more relaxed quantity than the safety gap in [6], which is defined under the optimal policy. Slater's constant can be from a feasible solution that is not necessarily optimal.

The next lemma shows that the optimal value of (10)-(12) is an upper bound on that of (1)-(2). The proof of this lemma can be found in [26].

**Lemma 1** *Assume $\{c(t)\}$ are i.i.d. across rounds, and $\{R(c,j)\}$ and $\{W^{(k)}(c,j)\}$ are i.i.d. samples given action $j$ and context $c$. Let $\pi^*$ be the optimal policy to problem (1)-(2) and $\mathbf{x}^*$ be the solution to (10)-(12) with entries $\{x_{c,j}^*\}_{c\in\mathcal{C},j\in[J]}$. We have*

$$\mathbb{E}\left[\sum_{t=1}^{T}\sum_{j\in[J]}R(c(t),j)X_j^{\pi^*}(t)\right] \leq T\sum_{c\in\mathcal{C},j\in[J]}p_c r(c,j)x_{c,j}^*.$$

The baseline problem (10)-(12) is the same as the one presented in Section 2 except that the tightness constant $\epsilon = 0$ here. Any feasible solution for the tightened problem in Section 2 is a feasible solution to the baseline problem. Under Slater's condition, when $\epsilon < \delta$, the tightened problem also has feasible solutions.

### 3.2 Regret and Constraint Violation Bounds

Given the baselines above, we now define regret and constraint violation.

**Regret:** Given policy $\pi$, we define the (pseudo)-regret of the policy to be

$$\mathcal{R}(\tau) = \tau\sum_{c\in\mathcal{C},j\in[J]}p_c r(c,j)x_{c,j}^* - \mathbb{E}\left[\sum_{t=1}^{\tau}\sum_{j\in[J]}R(c(t),j)X_j^{\pi}(t)\right]. \tag{13}$$

**Constraint violation:** The constraint violation in round $\tau$ is defined to be

$$\mathcal{V}(\tau) = \sum_{k\in[K]}\left(\mathbb{E}\left[\sum_{t=1}^{\tau}\sum_{j\in[J]}W^{(k)}(c(t),j)X_j^{\pi}(t)\right]\right)^{+}. \tag{14}$$

Note that the operator $(\cdot)^{+} = \max(\cdot, 0)$ is imposed so that different types of constraint violations will not be canceled out.

**Theorem 1** *Under Assumptions 1-3, the pessimistic-optimistic algorithm achieves the following regret and constraint violations bounds for any $\tau \in [T]$:*

$$\mathcal{R}(\tau) \leq \frac{60K^3}{\delta^3} + \frac{4\sqrt{6}K^{0.75}\sqrt{\tau}}{\delta} + 2 + \sqrt{8d\tau\beta_\tau(T^{-1})\log\left(\frac{d+\tau}{d}\right)},$$

$$\mathcal{V}(\tau) \leq K^{1.5}\left(\frac{48K^2}{\delta}\log\left(\frac{16}{\delta}\right) + \frac{24K^{1.5}}{\delta^2} + \frac{30K^{1.5}}{\delta} + 8K - \sqrt{\tau}\right)^{+}.$$

*where $\sqrt{\beta_\tau(T^{-1})} = m + \sqrt{2\log T + d\log(1 + \tau/d)}$.*

We make a few important observations from our theoretical results. First, for the *reward regret*, we observe

$$\mathcal{R}(\tau) = \tilde{\mathcal{O}}\left(\left(\frac{K^{0.75}}{\delta} + d\right)\sqrt{\tau}\right).$$

So the regret is independent of the number of contexts, action space $[J]$ and the dimension of cost functions $W^{(k)}(\cdot, \cdot)$. It grows sub-linearly in $\tau$ and the number of constraints $K$, and linearly in the dimension of reward feature $d$ and the inverse of Slater's constant $\delta$.

Second, for the *constraint violation*, we observe

$$\mathcal{V}(\tau) = \begin{cases} \tilde{\mathcal{O}}\left(\frac{K^{3.5}}{\delta} + \frac{K^3}{\delta^2}\right) & \tau \leqslant \tau' = \tilde{\mathcal{O}}\left(\frac{K^4}{\delta^2} + \frac{K^3}{\delta^4}\right) \\ 0 & \text{otherwise} \end{cases}. \tag{15}$$

That is, the constraint violation is *independent* of horizon $T$ and becomes *zero* when $\tau > \tau'$. The constraint violation, however, has a strong dependence on $K$ and $\delta$ when $\tau \leqslant \tau'$. This is not surprising because $K$ defines the number of constraints, and $\delta$ represents the tightness of the constraints (or size of the feasible set).

**Dependence on Slater's constant.** Both the regret and constraint violation increase in $\delta$. To see the intuition, note that $\delta$ determines the size of the feasible set for the optimization problem. A larger $\delta$ implies a larger feasible set, so it is easier to find a feasible solution, vice versa. Therefore, both regret and constraint violation increase as $\delta$ decreases because the problem becomes harder and requires more accurate learning.

**Sharpness of the bound.** In terms of horizon $T$, the bounds in Theorem 1 are sharp because the regret bound $\mathcal{R}(T)$ matches the instance-independent regret $\Omega(\sqrt{T})$ in multi-armed bandit problems without constraints [7, 18] up to logarithmic factors. Furthermore, zero constraint violation is the best possible. Therefore, the bounds are sharp up to logarithmic factors in terms of horizon $T$. It is not clear whether these bounds are sharp in terms of $K$, $d$, and $\delta$, which are interesting open questions.

### 3.3 A High Probability Bound on Constraint Violation

The constraint (2) defined in the original problem and the constraint violation measure defined in (14) are both in terms of expectation. An interesting, related question is what the probability is for a *sample-path version* of the constraints to be satisfied. It turns out that our algorithm provides a high probability guarantee on that as well. The proof can be found in [26].

**Corollary 1** *The pessimistic-optimistic algorithm guarantees that for any* $\tau \geqslant \frac{\kappa K^5}{\delta^2}\left(\log\left(\frac{K}{\delta}\right)\right)^2$, *where $\kappa$ is a positive constant independent of $\tau$, $K$, $\delta$ and $d$,*

$$\mathbb{P}\left(\sum_{t=1}^{\tau}\sum_{j\in[J]} W^{(k)}(c(t), j)X_j(t) > 0\right) = O\left(e^{-\frac{\delta\sqrt{\tau}}{50K^2}}\right).$$

## 4 Proof of Theorem 1

We first explain the intuition behind the main result. Recall that the algorithm selects action $j^*$ such that

$$j^* \in \arg\max_j\left(\hat{r}(c(\tau), j) - \frac{1}{V_\tau}\sum_k W^{(k)}(c(\tau), j)Q^{(k)}(\tau)\right),$$

and $V_\tau = O(\sqrt{\tau})$. Therefore, when $Q^{(k)}(\tau) = o(\sqrt{\tau})$, the reward term dominates the cost term, and our algorithm uses LinUCB to maximize the reward. When $Q^{(k)}(\tau) = \omega(\sqrt{\tau})$, the cost term dominates the reward term and our algorithm focuses on reducing $Q^{(k)}$. Slater's condition implies that there exists a policy that can reduce $Q^{(k)}$ by a constant (related to $\delta$) in each round. Therefore, the algorithm takes $\tilde{\mathcal{O}}(\sqrt{\tau})$ rounds to reduce $Q^{(k)}$ to $o(\sqrt{\tau})$, which may add $\tilde{\mathcal{O}}(\sqrt{\tau})$ to the regret

during this period. The argument above also implies that $Q^{(k)}(\tau) = O(\sqrt{\tau})$. Then, because

$$\mathbb{E}\left[\sum_{t=1}^{\tau}\sum_{j\in[J]} W^{(k)}(c(t),j)X_j(t)\right] \leqslant \mathbb{E}\left[Q^{(k)}(\tau+1)\right] - \sum_{t=1}^{\tau}\epsilon_t.$$

we can further bound the constraint violation at time $\tau$ to be a constant or even zero via the bound on $\mathbb{E}[Q^{(k)}(\tau+1)]$ and a proper choice of $\epsilon_t$.

## 4.1 Regret Bound

Now consider the regret defined in (13) and define $\mathbf{x}^{\epsilon_t,*}$ to be the optimal solution to the tightened problem (3)-(5) with $\epsilon = \epsilon_t$. We obtain the following decomposition by adding and subtracting corresponding terms:

$$\mathcal{R}(\tau) \overset{(a)}{=} \tau\sum_{c,j} p_c r(c,j)x^*_{c,j} - \mathbb{E}\left[\sum_{t=1}^{\tau}\sum_{j} r(c(t),j)X_j(t)\right]$$

$$= \underbrace{\sum_{t=1}^{\tau}\sum_{c,j} p_c r(c,j)\left(x^*_{c,j} - x^{\epsilon_t,*}_{c,j}\right)}_{\epsilon_t\text{-tight}} + \underbrace{\mathbb{E}\left[\sum_{t=1}^{\tau}\sum_{j}\left(r(c(t),j) - \hat{r}(c(t),j)\right)x^{\epsilon_t,*}_{c(t),j}\right]}_{\text{reward mismatch}}$$

$$+ \underbrace{\mathbb{E}\left[\sum_{t=1}^{\tau}\sum_{j}\left(\hat{r}(c(t),j)x^{\epsilon_t,*}_{c(t),j} - \hat{r}(c(t),j)X_j(t)\right)\right] - \sum_{t=1}^{\tau}\frac{K(1+\epsilon_t^2)}{V_t}}_{\text{Lyapunov drift}} \qquad (16)$$

$$+ \underbrace{\sum_{t=1}^{\tau}\frac{K(1+\epsilon_t^2)}{V_t}}_{\text{accumulated tightness}} + \underbrace{\mathbb{E}\left[\sum_{t=1}^{\tau}\sum_{j}\left(\hat{r}(c(t),j) - r(c(t),j)\right)X_j(t)\right]}_{\text{reward mismatch}}, \qquad (17)$$

where $(a)$ holds because the random reward is revealed after action $A(t)$ is taken so the noise is independent of the action. We next present a sequence of lemmas that bounds the terms above. The proofs of these lemmas are presented in [26].

First, we establish an upper bound on (16) by using the Lyapunov-drift analysis [31, 34]. We consider a Lyapunov function $L(t) = \sum_k Q^{(k)}(t)$ and show that maximizing the pseudo action value in our algorithm (**MaxValue** step) is equivalent to maximizing the following "reward minus Lyapunov drift"

$$V_t\sum_{j\in[J]}\hat{r}(c(t),j)X_j(t) - (L(t+1) - L(t)).$$

Therefore, our algorithm outperforms a static policy $x^{\epsilon_t}_{c(t),j}$ and results in the following lower bound

$$V_t\mathbb{E}\left[\sum_{j\in[J]}\hat{r}(c(t),j)X_j(t)\right] - \mathbb{E}[L(t+1) - L(t)] \geqslant V_t\sum_{j\in[J]}\hat{r}(c(t),j)x^{\epsilon_t}_{c(t),j} - K(1+\epsilon_t^2). \quad (18)$$

Apply the telescoping sum on (18) provides an upper bound on (16) in Lemma 2.

**Lemma 2** *Under the Pessimistic-Optimistic Algorithm, we have*

$$\mathbb{E}\left[\sum_{t=1}^{\tau}\sum_{j\in[J]}\hat{r}(c(t),j)\left(x^{\epsilon_t}_{c(t),j} - X_j(t)\right)\bigg|\mathbf{H}(t) = \mathbf{h}\right] - \sum_{t=1}^{\tau}\frac{K(1+\epsilon_t^2)}{V_t} \leqslant \sum_{t=1}^{\tau}\frac{Kt(\epsilon_t + \epsilon_t^2)\mathbb{I}(\epsilon_t > \delta)}{V_t}.$$

Next, we present Lemma 3 which establishes an upper bound on the $\epsilon_t$-tight term by comparing the optimal solution to the original baseline problem and that to its $\epsilon_t$−tightened version. It is based on the intuition that adding $\epsilon_t$-tightness to the feasible region only incurs $O(\epsilon_t)$ loss in the reward.

**Lemma 3** *Under Assumptions 1-3, we can bound the difference between the baseline optimization problem and its tightened version:*

$$\sum_{t=1}^{\tau}\sum_{c,j} p_c r(c,j)\left(x_{c,j}^* - x_{c,j}^{\epsilon_t,*}\right) \leqslant \sum_{t=1}^{\tau}\frac{\epsilon_t}{\delta}.$$

Finally, we present Lemma 4 that provides an upper bound on the reward mismatch terms. The proof is based on "self-normalized bound for vector-valued martingales" in [1] and shows that $\hat{r}(c,j)$ over-estimates $r(c,j)$ and converges to $r(c,j)$.

**Lemma 4** *Under the Pessimistic-Optimistic Algorithm, LinUCB guarantees that*

$$\mathbb{E}\left[\sum_{t=1}^{\tau}\sum_{j}(\hat{r}(c(t),j) - r(c(t),j))X_j(t)\right] \leqslant 1 + \sqrt{8d\tau\beta_\tau(T^{-1})\log\left(\frac{d+\tau}{d}\right)},$$

$$\mathbb{E}\left[\sum_{t=1}^{\tau}\sum_{j}(r(c(t),j) - \hat{r}(c(t),j))x_{c(t),j}^{\epsilon_t,*}\right] \leqslant 1.$$

Based on Lemmas 2, 3, and 4, we conclude that

$$\mathcal{R}(\tau) \leqslant \sum_{t=1}^{\tau}\frac{Kt(\epsilon_t + \epsilon_t^2)\mathbb{I}(\epsilon_t > \delta)}{V_t} + \sum_{t=1}^{\tau}\frac{\epsilon_t}{\delta} + \sum_{t=1}^{\tau}\frac{K(1+\epsilon_t^2)}{V_t} + 2 + \sqrt{8d\tau\beta_\tau(T^{-1})\log\left(\frac{d+\tau}{d}\right)}.$$

By choosing $\epsilon_t = K^{0.75}\sqrt{\frac{6}{t}}$ and $V_t = \delta K^{0.25}\sqrt{\frac{2t}{3}}$, we have

$$\sum_{t=1}^{\tau}\epsilon_t \leqslant 2K^{0.75}\sqrt{6\tau}, \quad \sum_{t=1}^{\tau}1/V_t \leqslant \frac{\sqrt{6\tau}}{K^{0.25}}, \quad \text{and} \quad \sum_{t=1}^{\tau}\frac{Kt(\epsilon_t + \epsilon_t^2)\mathbb{I}(\epsilon_t > \delta)}{V_t} \leqslant \frac{60K^3}{\delta^3},$$

which yields the regret bound

$$\mathcal{R}(\tau) \leqslant \frac{60K^3}{\delta^3} + \frac{4\sqrt{6}K^{0.75}\sqrt{\tau}}{\delta} + 2 + \sqrt{8d\tau\beta_\tau(T^{-1})\log\left(\frac{d+\tau}{d}\right)}.$$

## 4.2 Constraint Violation Bound

According to the dynamic defined in (9), we have

$$Q^{(k)}(\tau+1) \geqslant \sum_{t=1}^{\tau}\sum_{j}W^{(k)}(c(t),j)X_j(t) + \sum_{t=1}^{\tau}\epsilon_t,$$

where we used the fact $Q^{(k)}(0) = 0$. This implies the constraint violation can be bounded as follows:

$$\mathcal{V}(\tau) \leqslant \sum_{k}\left(\mathbb{E}[Q^{(k)}(\tau+1)] - \sum_{t=1}^{\tau}\epsilon_t\right)^+. \tag{19}$$

Next, we introduce a lemma on the upper bound of $Q^k(\tau)$. Define $\tau'$ the first time such that $\epsilon_{\tau'} \leqslant \delta/2$, that is, $\epsilon_\tau > \delta/2, \forall 1 \leqslant \tau < \tau'$.

**Lemma 5** *For any time $\tau \in [T]$ such that $\tau \geqslant \tau'$, i.e., $\epsilon_\tau \leqslant \delta/2$, we have*

$$\mathbb{E}\left[Q^{(k)}(\tau)\right] \leqslant \sqrt{K}\left(\frac{48K^2}{\delta}\log\left(\frac{16K}{\delta}\right) + 2K + \frac{4(V_\tau + K(1+\epsilon_\tau^2))}{\delta} + \tau' + \sqrt{K}\sum_{t=1}^{\tau'}\epsilon_t\right).$$

Based on our choices of $\epsilon_t = K^{0.75}\sqrt{\frac{6}{t}}$ and $V_t = \delta K^{0.25}\sqrt{\frac{2t}{3}}$, we obtain

$$\tau' = \frac{24K^{1.5}}{\delta^2} \quad \text{and} \quad \sum_{t=1}^{\tau}\frac{\epsilon_t}{\sqrt{K}} - \frac{4V_{\tau+1}}{\delta} \geqslant \sqrt{\tau} - 6K.$$

Note $\mathcal{V}(\tau) \leqslant K\tau'$ for any $\tau \leqslant \tau'$. Combine Lemma 5 into (19), we conclude

$$\mathcal{V}(\tau) \leqslant K^{1.5}\left(\frac{48K^2}{\delta}\log\left(\frac{16}{\delta}\right) + \frac{24K^{1.5}}{\delta^2} + \frac{30K^{1.5}}{\delta} + 8K - \sqrt{\tau}\right)^+.$$

# 5 Numerical Evaluations

In this section, we present numerical evaluations of the proposed algorithm, including 1) the constrained multi-armed bandit (MAB) example studied in [32]; 2) a constrained linear bandit example based on a healthcare dataset. We briefly report the setting and results due to the page limit. More details can be found in [26].

**The Constrained MAB Example in [32]:** To compare with OPA in [32], we studied the MAB example in [32] with $K = 4$-arms where the reward and cost distributions are Bernoulli with means $\bar{r} = (0.1, 0.2, 0.4, 0.7)$ and $\bar{c} = (0, 0.4, 0.5, 0.2)$ and the total cost in each round should not exceed $0.5$. The results are presented in Figure 2, where we can see that our algorithm has significant lower regret than that under OPA while the cost constraint is satisfied under both algorithms.

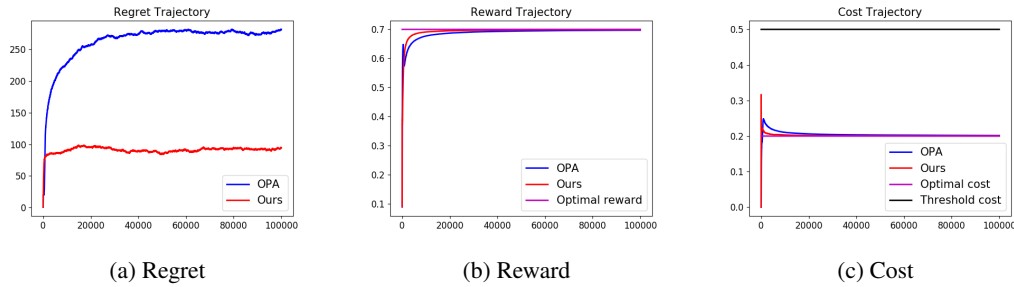

| (a) Regret | (b) Reward | (c) Cost |

Figure 2: Our Algorithm versus OPA [32]

**Constrained linear bandits for inpatient flow routing:** We also evaluated our algorithm with a real-world dataset on inpatient flow routing, where incoming patients have different features (context) such as age, gender, medical history, etc, and incur different amounts of "rewards" when being assigned to different wards (actions). We considered three types of constraints: capacity, fairness, and resource. We chose different learning horizons. The regrets and constraint violations at the end of the horizon are summarized in Table 1, which shows that our algorithm achieves a low regret and zero violation. This experiment also confirms that our algorithm guarantees the anytime cumulative constraints.

| $T$ | 2,500 | 10,000 | 22,500 | 40,000 | 62,500 |
|---|---|---|---|---|---|
| Regret | 24.38 | 51.89 | 74.38 | 90.39 | 106.82 |
| Largest constraint violation of the three types | 0 | 0 | 0 | 0 | 0 |

Table 1: Regret and Constraint Violations under Our Algorithm for Inpatient Flow Routing

# 6 Conclusions and Extensions

In this paper, we study stochastic linear bandits with general anytime cumulative constraints. We develop a pessimistic-optimistic algorithm that is computationally efficient and has strong guarantees on both regret and constraint violations. We conclude this paper by mentioning an extension on the case where the cost signals $W^{(k)}(c(t), A(t))$ are revealed after action $A(t)$ is taken. However, we assume the costs can be linearly parameterized as in [32], i.e. $W(c(t), A(t)) = \langle \mu_*, \psi(c, j) \rangle + \xi(t)$, where $\psi(c, j) \in \mathbb{R}^d$ is a feature vector, $\mu_* \in \mathbb{R}^d$ is an unknown underlying vector, and $\xi(t)$ is a zero-mean random variable. In this case, we also obtain an estimate $\breve{W}(c(t), j)$ of $W(c(t), j)$ with LinUCB and replace $W(c(t), j)$ with $\breve{W}(c(t), j)$ in the steps of MaxValue and Dual Update in the Pessimisitic-Optimistic Algorithm. This variation of Pessimisitic-Optimistic Algorithm has a similar regret and constraint violation guarantees in Theorem 2 (the details are in [26]).

**Theorem 2 (Informal)** *With linear costs as in [32], a variation of our algorithm achieves* $\mathcal{R}(\tau) = \tilde{\mathcal{O}}\left(\frac{d}{\delta}\sqrt{\tau} + \frac{d^4}{\delta^3}\right)$ *for any* $\tau \in [T]$ *and* $\mathcal{V}(\tau) = 0$ *for* $\tau \geqslant \tau'' = O\left(\frac{d^2}{\delta^4}\log^2 T\right)$.

**Acknowledgment:** This work has been supported in part by NSF CNS-2001687, CNS-2002608 and CNS-2152657.

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
