# OpenReview forum: "An Efficient Pessimistic-Optimistic Algorithm for Stochastic Linear Bandits with General Constraints"
_NeurIPS.cc/2021/Conference — NeurIPS 2021 Poster_

### Official Review · Reviewer_WupD · 2021-07-01

**Rating:** 7
**Confidence:** 3

**Summary:**

This paper proposes a learning algorithm for a stochastic linear bandits with constraints. At each decision step, the decision maker can choose between J actions. Each action A_t then provides a context-dependent reward $r(C_t, A_t)$ and contributes a $W_k(C_t, A_t)$ to the "constraint k". The goal of the learner is to maximize the reward provided that the K constraint $\sum_{t=0}^\tau W_k(C_t, A_t) \le 0$ are respected.

The authors present an algorithm that guarantees a sublinear regret in $O(\sqrt{T})$ while respecting constraints for all time-steps except at most O(1) time-steps.

After presenting the algorithm, most of the paper is devoted to a careful analysis of the theoretical properties of the algorithm. No application nor experimental results are presented.


**Ethical Concerns:**

NA.

**Limitations And Societal Impact:**

There is no direct societal impact foreseen.

The theoretical part of the paper is strong but there is not empirical evaluation of the algorithm. This could be further discussed in the paper (the only discussion for now is to say that "it can be implemented easily").


**Main Review:**

I think that the problem studied in the paper is very relevant. Most of the learning algorithms are either presented in a unconstraint setting or in very restrictive setting (like anytime action constraint). The type of constraints chosen in this paper are both quite expressive (they can represent fairness, budget,...) and also quite flexible to allow for efficient algorithms.

The algorithm presented in the paper is convincing. The algorithm is based on a primal-dual approach and uses virtual queues to estimate the Lagrange multipliers. I did not check all details but the proof methodology looks sound: The analysis is based on an optimistic algorithm (LinUCB) plus a Lyapunov-drift method.

The performance guarantee of the algorithm seem very good as the algorithm can provide a O(T) regret with a O(1) constraint violation.

From a methodological point of view, the paper is very close to the recent paper "POND: Pessimistic-Optimistic oNline Dispatching" by Liu et al.  Apart from the struct of the reward, the two settings are very close.

To me, the main limitation of the paper is that it is purely focussed on theory and does not provide any experimental algorithm. One may wonder if this is to keep the paper more consistant or because:
- formulating a real problem in this setting is difficult?
- the hidden constants in the O(.) are too large to make the algorith practical?

Also, the reward setting with finite action space and linear payoff is quite specific and closely related to bandits. Would the approach be generalizable to more general contextual bandit settings?


**Time Spent Reviewing:**

2

---

> ### Author Response · Authors · 2021-08-10
> **Response to Reviewer WupD**
>
> We greatly appreciate the encouraging comments from the reviewer. We want to mention that POND is an unpublished arXiv paper.
>
> We only included theoretical results in the paper to emphasize our theoretical contributions. Our algorithm in fact is very practical with important applications. We have evaluated the proposed algorithm using a real-world dataset in healthcare on inpatient flow routing, which is also an important motivating application. We consider routing patients to different inpatient wards. Patients come with different features (context) such as age, gender, medical specialty, etc, and incur different “reward” when assigned to different wards, corresponding to different actions. The reward is different because the level of care provided by different wards match with the patient needs differently, and we measure the reward via the avoided 30-day readmission penalty, i.e., a reward is collected if the patient is not readmitted to the hospital within 30 days since being discharged.
>
> Calibrating from the data we have, we formulate three types of constraints: capacity, fairness, and resource. After normalizing, the capacity constraint for each ward is $[0.2, 0.2, 0.175, 0.175, 0.175, 0.175];$ the fairness requirement is $[0.175, 0.175, 0.15, 0.15, 0.125, 0.125];$ and the nursing resource constraint is $[0.1875, 0.1875, 0.1875, 0.1875, 0.1875, 0.1875]$, where each patient consumes "one" unit after being assigned to that server. We note that these constraints are strict since the hospital capacity is highly constrained.
>
> In the simulation, we set scaling parameter $V_t = 4\sqrt{t}$ and slackness parameter $\epsilon_t = 1/\sqrt{t}$ in our proposed algorithm.  We run $T=[2500, 10000, 22500, 40000, 62500].$ In all experiments, we report the regret and constraint violations at the end of learning horizon $T$ (we report the maximum violation values out of each type of constraints) in following table.
>
> |  $T$   | 2500 | 10000 | 22500 | 40000 | 62500 |
> |  :--  | :--:  | :--:  | :--: | :--: |  :--: |
> | Regret | 24.38 | 51.89 | 74.38 | 90.39 | 106.82 |
> | Capacity Violation | 0 | 0 | 0 | 0 | 0 |
> | Fairness Violation | 0 | 0 | 0 | 0 | 0 |
> | Resource Violation | 0 | 0 | 0 | 0 | 0 |
>
> To further justify anytime constraint violation, we extracted the trajectory with $T=10000$ to see how the violations evolve and if zero constraint violation can be obtained after constant $\tau'$ round. The results justify our theory and we listed $\tau'$ for each type of constraints in the following table.
>
> | Constraint Types | Capacity | Fairness | Resource |
> |  :--: | :--:  | :--:  | :--: |
> | $\tau'$ | 32 | 50 | 54 |
>
> We will include these experimental results (and more) in the complete version of the paper.

---

> > ### Comment · Reviewer_WupD · 2021-08-31
> > **Post-rebuttal**
> >
> > I thank the authors for their answer. I maintain my score of 7.

---

> > > ### Author Response · Authors · 2021-08-31
> > > **Response to Reviewer WupD**
> > >
> > > We sincerely thank the reviewer for the positive feedback!

---

### Official Review · Reviewer_1dmt · 2021-07-15

**Rating:** 7
**Confidence:** 3

**Summary:**

This paper proposes an efficient pessimistic-optimistic algorithm for stochastic linear bandits with general nonlinear constraints. The authors give a later-constant-dependent regret that, compared to the unconstrained case, achieves a similar complexity order in terms of T but higher-order dependence on d. This algorithm also achieves zero-constraints violation after a certain time \tau, where $\tau$ is independent.  Besides the main results, the author also gives several side results including a high probability bound on sample-path version of the constraints and abound for linear cost when the cost is revealed after the learning taking actions (so it's also bandits).

**Limitations And Societal Impact:**

The authors didn't address the limitation of their work. From a theoretical perspective, I think there are no big limitations. One concern I am thinking about is whether such an algorithm can achieve some instance-dependent regret as I raised in the main review.

**Main Review:**

--Originality:
While the analysis on the optimism part is standard LinUCB, the techniques the author uses to adaptively estimate the Lagrange multipliers via tracking its violations and choosing proper scaling parameters are novel. Also, this work makes an interesting connection between Slater’s constant and regret/violation updates.

--Quality:
This submission is technically sound, all the claims are supported.

--Clarity:
This paper is well-written with clear intuition explanation and rigorous proof.

--Significance:
The results are meaningful and some analysis techniques including dual updates and the connection to Slater's constant are inspiring.

--Question:
I wonder if this algorithm could be directly extended to getting instance-dependent bound? In particular, I know linUCB is unable to achieve the optimal instance-dependent bound, but how about the bound depends on, for example, $\min_{c}  \min_{x’} \max_x <\theta, \phi(c,x) - \phi(c,x’)> $ ? Because I notice that, there are many parts in analysis that greedily use the term \sqrt{\tau} (scaling parameter V, budget $\epsilon_t$ ). It seems that aiming at $\sqrt{T}$ makes the whole problem less challenging.



**Time Spent Reviewing:**

6

---

> ### Author Response · Authors · 2021-08-10
> **Response to Reviewer 1dmt**
>
> We first would like to sincerely thank the reviewer for the very encouraging comments. About instance-dependent bounds, it is indeed a much more challenging problem, which we are currently looking at. With the constraints, the techniques used for obtaining instance-dependent bounds for unconstrained bandit problems do not directly apply.

---

### Official Review · Reviewer_DVmW · 2021-07-15

**Rating:** 6
**Confidence:** 4

**Summary:**

This paper considers stochastic linear bandits with general nonlinear constraints. They propose a pessimistic-optimistic algorithm for this problem, which obtains a sublinear of time and is computationally efficient.

**Limitations And Societal Impact:**

No, the authors have not addressed the limitations. Please see my comments in the main review.

**Main Review:**

The paper is well written and easy to follow. I think that the main contribution of this paper is to extend linear bandits with linear constraints in [6], [32] to the ones with general nonlinear constraints.  However, there are several main limitations that make me lean towards rejection.
(1)	The authors said that their results in a “much more efficient calculation” comparing to OPLB proposed in [32]. However, OPLB is designed for contextual bandit which is more general than the problem this paper is considering. More importantly, [32] also proposed an algorithm for linear bandits, called OPB, which can be “implemented extremely efficiently” as [32] mentioned. Thus, it would be fair if the authors compare their algorithm to OPB.
(2)	Because the authors propose a new algorithm, experimental results are needed to demonstrate the performance compared to existing algorithms such as OPB in [32], SLUCB in [6].
(3)	The idea of using a pessimistic-optimistic mechanism has been exploited in [32].  The primal-dual approach in optimization may be different from [32], however, this novelty is not enough so that it is a theoretical paper.


**Time Spent Reviewing:**

3

---

> ### Author Response · Authors · 2021-08-10
> **Response to Reviewer DVmW**
>
> **Comparison with [32]:** We would like to respectfully point out that the reviewer misunderstood [32]. OPLB is for stochastic *linear* bandits, not for general contextual bandits. The computationally efficient algorithm, OPB, is for the traditional multi-armed bandits (MAB), not for linear bandits.  Therefore, it is indeed fair to compare our algorithm with OPLB, instead of OPB. As the authors themselves acknowledged on page 5 in [32] that the main bottleneck of OPLB is the computational complexity, which makes it challenging to implement it in practice. The computational complexity of our algorithm, on the other hand, is similar to LinUCB, and can be easily implemented.
>
> **Pessimistic-optimistic approach:** The pessimistic-optimistic approach is a high-level idea. The algorithm and the analysis used in this paper are fundamentally different from [32]. For example, the pessimism is achieved via adding a time-varying tightness constant, which is completely different from the pessimism in [32].
>
> **Experimental result:** We have evaluated the proposed algorithm using a real-world dataset in healthcare on inpatient flow routing, which is also an important motivating application. We consider routing patients to different inpatient wards. Patients come with different features (context) such as age, gender, medical specialty, etc, and incur different “reward” when assigned to different wards, corresponding to different actions. The reward is different because the level of care provided by different wards match with the patient needs differently, and we measure the reward via the avoided 30-day readmission penalty, i.e., a reward is collected if the patient is not readmitted to the hospital within 30 days since being discharged.
>
> Calibrating from the data we have, we formulate three types of constraints: capacity, fairness, and resource. After normalizing, the capacity constraint for each ward is $[0.2, 0.2, 0.175, 0.175, 0.175, 0.175];$ the fairness requirement is $[0.175, 0.175, 0.15, 0.15, 0.125, 0.125];$ and the nursing resource constraint is $[0.1875, 0.1875, 0.1875, 0.1875, 0.1875, 0.1875]$, where each patient consumes “one” unit after being assigned to that server. We note that these constraints are strict since the hospital capacity is highly constrained.
>
> In the simulation, we set scaling parameter $V_t = 4\sqrt{t}$ and slackness parameter $\epsilon_t = 1/\sqrt{t}$ in our proposed algorithm.  We run $T=[2500, 10000, 22500, 40000, 62500].$ In all experiments, we report the regret and constraint violations at the end of learning horizon $T$ (we report the maximum violation values out of each type of constraints) in following table.
>
> |  $T$   | 2500 | 10000 | 22500 | 40000 | 62500 |
> |  :--  | :--:  | :--:  | :--: | :--: |  :--: |
> | Regret | 24.38 | 51.89 | 74.38 | 90.39 | 106.82 |
> | Capacity Violation | 0 | 0 | 0 | 0 | 0 |
> | Fairness Violation | 0 | 0 | 0 | 0 | 0 |
> | Resource Violation | 0 | 0 | 0 | 0 | 0 |
>
> To further justify anytime constraint violation, we extracted the trajectory with $T=10000$ to see how the violations evolve and if zero constraint violation can be obtained after constant $\tau'$ round. The results justify our theory and we listed $\tau'$ for each type of constraints in the following table.
>
> | Constraint Types | Capacity | Fairness | Resource |
> |  :--: | :--:  | :--:  | :--: |
> | $\tau'$ | 32 | 50 | 54 |
>
> We will include these experimental results (and more) in the complete version of the paper.

---

> > ### Comment · Reviewer_DVmW · 2021-09-02
> > **Increasing my score**
> >
> > Thank you for your response. I agree with the authors that my concern about the OPB algorithm is inaccurate. I would like to increase my score to 6. However, on my concern about experiments, I encourage the authors to incorporate additional experiments in comparison with the related algorithms such as SLUCB, GSLUCB.

---

> > > ### Author Response · Authors · 2021-09-02
> > > **Thanks!**
> > >
> > > We sincerely thank the reviewer for considering our response and revising the score! We will definitely look into SLUCB, GSLUCB, and other related algorithms, and will add more experimental results in the revision.

---

### Official Review · Reviewer_1DcG · 2021-07-15

**Rating:** 7
**Confidence:** 4

**Summary:**

This paper studies stochastic linear bandits with anytime cumulative constraints on (non)linear costs. It proves a sublinear regret bound for the reward, and beyond round $\tau'$, it shows that the proposed primal-dual algorithm has zero constraint violation.

**Main Review:**

*Main Comments*
- The analysis and proof structure is well-written and easily understandable. I have verified the details and am satisfied with the correctness of the proof.
- The observation model in line 128 for the constraints is the full-information setting,  i.e. the learner observes the costs $W^k(c(t),  j)$ for each $j$, and not just the cost of the played arm. Please justify this feedback model. This also needs to be made explicit in the introduction. This is different compared to [32] which observes the cost only of the played arm, and perhaps the comparison to [32] is not fair.
- In Section 5, the paper also considers linear cost functions and obtains similar regret guarantees (by replacing the empirical estimates with the UCB estimates). Again, this needs to be explicitly explained in the introduction. Since this setting is now the same as [32], please compare the technical novelty in the analysis and the bounds to those obtained in [32].

*Details*
- It would be helpful to give examples that require anytime cumulative constraints, and which are not covered by cumulative constraints (not anytime) or anytime policy constraints. The setting studied by the paper needs to be better motivated.
- How does the algorithm set the Slater constant $\delta$? Is the safe action known to the algorithm in advance?
- It would be helpful to do some synthetic experiments to verify the correctness of the algorithm and the regret bound. Of particular interest would be to empirically verify that there is no constraint violation after $\tau^\prime$ rounds.

**Time Spent Reviewing:**

5

---

> ### Author Response · Authors · 2021-08-10
> **Response to Reviewer 1DcG**
>
> We would like to sincerely thank the reviewer for the encouraging comments!
>
> **Observation Model:** One important application of the observation model is fairness constraints in linear bandits, where each action represents a worker and each worker should receive at least $\rho_k$ fraction of the workload. In this case, the learner knows the costs (the workload). We will clarify this in the introduction.
>
> **Comparison with [32]:** When the cost function is linear, our algorithm has similar regret with that in [32] and guarantees zero constraint violation for large $\tau$. The key advantage of our algorithm compared with [32] is the computational complexity.  As the authors of [32] themselves acknowledged on page 5 in the arXiv version that the main bottleneck of OPLB is the computational complexity, which makes it challenging to implement in practice. The computational complexity of our algorithm, on the other hand, is similar to LinUCB, and can be easily implemented. We discussed this on page 2 of our paper, but will further elaborate in the introduction. Our analysis for violation constraints based on the Lyapunov drift analysis is of independent interest.
>
> **Estimate of $\delta$:** One possible way is to estimate $\delta$ by using the safe action as in [6,32].
>
> **Evaluation:** We have evaluated the proposed algorithm using a real-world dataset in healthcare on inpatient flow routing, which is also an important motivating application. We consider routing patients to different inpatient wards. Patients come with different features (context) such as age, gender, medical specialty, etc, and incur different "reward" when assigned to different wards, corresponding to different actions. The reward is different because the level of care provided by different wards match with the patient needs differently, and we measure the reward via the avoided 30-day readmission penalty, i.e., a reward is collected if the patient is not readmitted to the hospital within 30 days since being discharged.
>
> Calibrating from the data we have, we formulate three types of constraints: capacity, fairness, and resource. After normalizing, the capacity constraint for each ward is $[0.2, 0.2, 0.175, 0.175, 0.175, 0.175];$ the fairness requirement is $[0.175, 0.175, 0.15, 0.15, 0.125, 0.125];$ and the nursing resource constraint is $[0.1875, 0.1875, 0.1875, 0.1875, 0.1875, 0.1875]$, where each patient consumes "one" unit after being assigned to that server. We note that these constraints are strict since the hospital capacity is highly constrained.
>
> In the simulation, we set scaling parameter $V_t = 4\sqrt{t}$ and slackness parameter $\epsilon_t = 1/\sqrt{t}$ in our proposed algorithm.  We run $T=[2500, 10000, 22500, 40000, 62500].$ In all experiments, we report the regret and constraint violations at the end of learning horizon $T$ (we report the maximum violation values out of each type of constraints) in following table.
>
> |  $T$   | 2500 | 10000 | 22500 | 40000 | 62500 |
> |  :--  | :--:  | :--:  | :--: | :--: |  :--: |
> | Regret | 24.38 | 51.89 | 74.38 | 90.39 | 106.82 |
> | Capacity Violation | 0 | 0 | 0 | 0 | 0 |
> | Fairness Violation | 0 | 0 | 0 | 0 | 0 |
> | Resource Violation | 0 | 0 | 0 | 0 | 0 |
>
> To further justify anytime constraint violation, we extracted the trajectory with $T=10000$ to see how the violations evolve and if zero constraint violation can be obtained after constant $\tau'$ round. The results justify our theory and we listed $\tau'$ for each type of constraints in the following table.
>
> | Constraint Types | Capacity | Fairness | Resource |
> |  :--: | :--:  | :--:  | :--: |
> | $\tau'$ | 32 | 50 | 54 |
>
> We will include these experimental results (and more) in the complete version of the paper.

---

> > ### Comment · Reviewer_1DcG · 2021-08-30
> > **Post-rebuttal**
> >
> > I thank the authors for their response. I have gone through the rebuttal and the other reviews and would like to maintain my score of 7.
> >
> > I encourage the authors to incorporate the comments from the reviews (better motivation and examples of the problem setting, comparison with [32], explicitly differentiate between the two observation models) and include the experimental results in the next version of the paper.

---

> > > ### Author Response · Authors · 2021-08-31
> > > **Response to Reviewer 1DcG**
> > >
> > > We sincerely thank the reviewer for the great comments and suggestions. We will definitely incorporate them in our future version.

---

### Official Review · Reviewer_nU53 · 2021-07-24

**Rating:** 5
**Confidence:** 4

**Summary:**

The paper considers stochastic linear bandits with (possibly nonlinear) cumulative constraints that need to be satisfied on expectation. For this problem they propose a pessimistic-optimistic algorithm with the following properties: (i) it has O(\sqrt(\tau)) regret at each round \tau; (ii) it has zero constraint violations at any round \tau>\tau' for a constant number of rounds \tau'.

**Limitations And Societal Impact:**

Limitations of the model/setting/algorithm/analysis are not clearly discussed

**Main Review:**

The paper has potentially some interesting ideas. However, I am afraid that I am leaning towards recommending rejection in its current form for the following main reasons:

(i) Insufficient reference to related works.
First, in my opinion it is not good practice to defer the entire section on related works to the appendix. Second, the authors have missed citing a series of recent related works on safe bandits; for example (and several more...)
* Khezeli, K. and Bitar, E. (2019). Safe linear stochastic bandits. arXiv preprint arXiv:1911.09501
* Kazerouni, A., Ghavamzadeh, M., Abbasi, Y., and Van Roy, B. (2017). Conservative contextual
linear bandits. In Guyon, I., Luxburg, U. V., Bengio, S., Wallach, H., Fergus, R., Vishwanathan, S.,
and Garnett, R., editors, Advances in Neural Information Processing Systems 30, pages 3910–3919.
Curran Associates, Inc.
* Moradipari, Ahmadreza, Christos Thrampoulidis, and Mahnoosh Alizadeh. "Stage-wise Conservative Linear Bandits." Advances in Neural Information Processing Systems 33 (2020).
* HasanzadeZonuzy, Aria, Dileep M. Kalathil, and Srinivas Shakkottai. "Learning with safety constraints: Sample complexity of reinforcement learning for constrained mdps." arXiv preprint arXiv:2008.00311 (2020)
* Moradipari, Ahmadreza, et al. "Safe linear thompson sampling." arXiv preprint arXiv:1911.02156 (2019).

(ii) Comparison to related works.
The authors claim that their setting is more relaxed than that of [32]. But, they do not seem to discuss what their result implies specifically for the setting of [32] and how exactly it compares to their result.
Lines 81-82: Constraints in [6] hold with high-probability, but they are *not* on expectation as in this paper's setting. Thus, the claim does not appear correct.

(iii) Motivation.
What is the motivation behind the formulation in (2)? Can you provide concrete examples elaborating on the potential applications on "safety, fairness, and budget constraints"?

In addition to the points above:
(iv) Simulation results might help the reader better understand the algorithm's operation
(v) How is \delta estimated to be used in the algorithm? How does the mismatch in estimating this quantity affects the regret and constraint violations?
(vi) Intuitively knowledge that the constraints have specific structure (eg linear) should affect performance. How is this reflected in the bounds?


===============
Increase score to 5 after rebuttal. See discussion.

**Time Spent Reviewing:**

4

---

> ### Author Response · Authors · 2021-08-10
> **Response to Reviewer nU53**
>
> While we greatly appreciate the reviewer's comments, we respectfully disagree with the rating. The three main reasons mentioned by the reviewer are about related work and motivation. These comments can be easily addressed with minor changes; and in fact, the comparison with [32] is already in the paper. Therefore, we believe these should not be decisive factors in judging the paper. The rating should be judged based on its novelty and contributions. We next provide a detailed response to the three main comments. We sincerely hope the reviewer will reconsider the rating based on our response.
>
> **Insufficient Reference:**
> The current version includes a brief discussion on the most related work in the introduction and defers the detailed discussion to the appendix. This is due to the page limit and the feedback we received from the COLT submission where the reviewer suggested us to just keep the most important ones in the main submission so that the paper can focus more on explaining the main results. With the 9-page limit, we feel it is a good compromise. We will certainly include the related work in the main paper (after introduction) in the full version of the paper.
>
> About the additional references suggested by the reviewer, we would be happy to include them in the revision. However, we would like respectively point out that these references are not the ones most related to this paper.  [1], [2], and [3] consider “conservative” bandits, where the “conservative” constraint has a very special structure and is imposed directly on the reward. We consider general constraints that are defined independently from the reward. [4] studied episodic constrained reinforcement learning in the *tabular setting* instead of linear bandits. The tabular setting is like the traditional MAB and cannot be applied to linear approximation (linear bandits). [5] is related to the paper "Linear Stochastic Bandits under Safety Constraints" which we discussed and compared with. Both of the papers consider anytime action constraints, different from anytime cumulative constraints in this paper.
>
>
> **Comparison with [32] and [6]:** Our setting applies to general (possibly nonlinear) cost functions, so is more general than [32]. We also explicitly considered the linear cost model studied in [32] in Theorem 2 (the details are included in the supplemental material), where we show that our algorithm achieves a similar regret bound and zero constraint violation for large $\tau.$ More importantly, our algorithm does not need to solve a constrained optimization problem at each step, so is computationally efficient. The authors of [32] themselves mention on page 5 of their arXiv version that the major bottleneck of their algorithm, OPLB, is its computational complexity. The computationally efficient algorithm OPB only works for MAB, not for linear bandits studied in our paper. The discussion on computational complexity is on page 2 of the current version. We will expand this discussion according to the reviewer's comment.
>
>
>
> Constraints in [6] is also defined on the *expected cost* of the action at time $t,$ not the actual realization of the cost (see the cost constraint (1) on page 2 of [6] where the noise $\eta_t$ does not appear in the constraint). So the high probability result means the expectation is less than $c$ with a high probability. Our result states that the expectation constraint holds with probability one, which is stronger than the high probability result in [6].
>
>
>
> **Motivation:** By setting different forms of $W$, our constraint includes several important constraints. Consider the example that we have $K$ servers and action $k$ is to use server $k$ to serve the incoming context.
>
> - When $W^{(k)}(c(t), A(t))=\rho_k- \mathbb{I}_{A(t)=k},$ the constraint can represent a fairness constraint such that server $k$ should be used to serve at least $\rho_k$ fraction of the jobs.
>
> - When $W^{(k)}(c(t), A(t))=w_k\mathbb{I}_{A(t)=k}-b_k,$  it can represent a budget constraint such that it costs $w_k$ \\$ to serve a job at server $k$ and we can spend at most $b_k$ \\$ on average on server $k.$
>
> We thank the reviewer for this comment and will include these examples in the revision.
>
> **Simulation:** We evaluated our algorithm in a healthcare application with a real hospital dataset. We consider routing patients to different inpatient wards. Patients come with different features (context) such as age, gender, medical specialty, etc, and incur different "reward" when assigned to different wards, corresponding to different actions. The reward is different because the level of care provided by different wards match with the patient needs differently, and we measure the reward via the avoided 30-day readmission penalty, i.e., a reward is collected if the patient is not readmitted to the hospital within 30 days since being discharged.
>
> Calibrating from the data we have, we formulate three types of constraints: capacity, fairness, and resource. After normalizing, the capacity constraint for each ward is $[0.2, 0.2, 0.175, 0.175, 0.175, 0.175];$ the fairness requirement is $[0.175, 0.175, 0.15, 0.15, 0.125, 0.125];$ and the nursing resource constraint is $[0.1875, 0.1875, 0.1875, 0.1875, 0.1875, 0.1875]$, where each patient consumes ``one" unit after being assigned to that server. We note that these constraints are strict since the hospital capacity is highly constrained.
>
> In the simulation, we set scaling parameter $V_t = 4\sqrt{t}$ and slackness parameter $\epsilon_t = 1/\sqrt{t}$ in our proposed algorithm.  We run $T=[2500, 10000, 22500, 40000, 62500].$ In all experiments, we report the regret and constraint violations at the end of learning horizon $T$ (we report the maximum violation values out of each type of constraints) in following table.
>
> |  $T$   | 2500 | 10000 | 22500 | 40000 | 62500 |
> |  :--  | :--:  | :--:  | :--: | :--: |  :--: |
> | Regret | 24.38 | 51.89 | 74.38 | 90.39 | 106.82 |
> | Capacity Violation | 0 | 0 | 0 | 0 | 0 |
> | Fairness Violation | 0 | 0 | 0 | 0 | 0 |
> | Resource Violation | 0 | 0 | 0 | 0 | 0 |
>
> To further justify anytime constraint violation, we extracted the trajectory with $T=10000$ to see how the violations evolve and if zero constraint violation can be obtained after constant $\tau'$ round. The results justify our theory and we listed $\tau'$ for each type of constraints in the following table.
>
> | Constraint Types | Capacity | Fairness | Resource |
> |  :--: | :--:  | :--:  | :--: |
> | $\tau'$ | 32 | 50 | 54 |
>
> We will include these experimental results (and more) in the complete version of the paper.

---

> > ### Comment · Reviewer_nU53 · 2021-09-03
> > **Re**
> >
> > Thank you for your response. I apologize for the delay.
> >
> > **Re:** “we would like respectively point out that these references are not the ones most related to this paper.
> > Please allow me to disagree. In my humble opinion, work that has motivated your “most related works” [6,32] is related and should be referenced.
> >
> > [2] also considers cumulative constraints as in your work.
> >
> > Re: “[5] is related to the paper "Linear Stochastic Bandits under Safety Constraints" which we discussed and compared with. Both of the papers consider anytime action constraints, different from anytime cumulative constraints in this paper.”
> > So after all it is related work  If anything, compared to [6], in [5] the constraints do not involve the reward similar to [32] and your work.  Please also see Section 7 in [32].
> >
> > In my opinion all these works are related and should be discussed. Also, again IMO proper reference/discussion to related work is important aspect of the paper.
> >
> > **Re:** “Our setting applies to general (possibly nonlinear) cost functions, so is more general than [32]. “
> > 1.	As the authors mention in the intro [32] considers anytime constraints rather than anytime policy constraints. Since these are not the same (as the authors say in line 50 they are “similar” but not same), the results are not immediately comparable.
> > 2.	I looked at Section I. The regret seems to have an unfavorable dependency on dimension d (fourth power) and \delta (third power) unlike the result in [32]. This is not discussed in the paper.
> > 3.	Your Assumption 3(Slater’s condition) does not seem to be required in [32]. You mention that it is needed in [6], but [6] considers a different setting where constraint is same as reward.
> >
> > **Re:** Simulation result
> >
> > Are all the constraints linear in your simulation? What is the dimension d here? How do you set \delta?
> > Have you compared to any baselines?

---

> > > ### Author Response · Authors · 2021-09-03
> > > **Response to Reviewer nU53**
> > >
> > > Not a problem at all. We sincerely thank your response to our response! We completely agree that reference/discussion on related work is important, and we greatly appreciate the additional references and additional differences with other references (e.g. [32]) you pointed out. As we said in the response, we will definitely include these additional references with further discussions. What we wanted to say is that it is a bit unusual to recommend rejection because of missing a few references. It is something that can be easily addressed with minor changes so is often suggested as a minor revision. We would be very grateful if the reviewer can revisit the score based on our commitment to including and expanding the related work discussion as you suggested.
> > >
> > > We also would like to say a few things about [32]: our paper and [32] do consider different types of constraints. What we were saying in the response is that the cost function in our constraint is more general than [32] and we are not claiming our constraint is more general. The key contribution of our paper with respect to [32] is the computational complexity. As the authors themselves acknowledged on page 5 in [32] that the main bottleneck of OPLB is the computational complexity, which makes it difficult to implement it in practice. The computational complexity of our algorithm, on the other hand, is similar to LinUCB, and can be easily implemented.
> > >
> > > In terms of simulations:  Yes, all the constraints are linear in our simulation. The dimension of d is 10 including (age, gender, length of stay, etc.). We set $\delta = 0.1$. In terms of baseline, the algorithm that is mostly related is OPLB [32], whose computational complexity, however, is too high for any practical implementation (as the authors of [32] themselves acknowledged in their paper). SLUCB and GSLUCB [6] are two other possible baselines, but they assume the constraints are deterministic (but unknown) so cannot be used in the current experiment setting. We are currently working on settings where we can compare our algorithm with SLUCB, GSLUCB, and OPLB. The current experiment shows the significance of our algorithm, which can be used to efficiently solve a problem to which existing algorithms SLUCB, GSLUCB, OPLB cannot be even applied. But we will identify other settings where ours can be compared with existing algorithms.

---

> > > > ### Comment · Reviewer_nU53 · 2021-09-03
> > > > **Thank you for your quick response**
> > > >
> > > > Thank you for your quick response.
> > > >
> > > > **Re:**"because of missing a few references."
> > > >
> > > > Not to get into an endless loop here, but to me you are missing important references. When the authors are not familiar with these closely related work, it is harder for me as a reviewer (that given the time constraints/limitations of NeurIPS review process) to be 100% convinced about the technical & algorithmic novelty and implication of the work.
> > > >
> > > > **Related:** "The current experiment shows the significance of our algorithm, which can be used to efficiently solve a problem to which existing algorithms SLUCB, GSLUCB, OPLB cannot be even applied."
> > > > Sure, they are not applicable because they are not designed for type of constraints you mention, but could there be natural adjustments to them so that they are applicable in this setting? Again, doing that exercise is not taking away from your contribution. IMO it would emphasize it.
> > > > I encourage you to look more carefully at those references and perhaps a few other related ones (eg what about the Safe-OPT and Stage-OPT algorithms designed for safe GP optimization?)
> > > >
> > > > **Re**: our paper and [32] do consider different types of constraints. What we were saying in the response is that the cost function in our constraint is more general than [32] and we are not claiming our constraint is more general.
> > > >
> > > > This should be more clearly stated in the paper and when comparing results to [32]. Please also include the discussion about differences in scalings of d, \delta between your algorithms when presenting Thm 2 in the last section.
> > > >
> > > > **Re**: "The key contribution of our paper with respect to [32] is the computational complexity."
> > > >
> > > > But... their setting is not the same. How critical is that in allowing your algorithm/proof to work? Do you think your technique could apply to the setting of [32]?
> > > > I understand it is late in the discussion process and is my fault so I do not necessarily expect an answer here, but would be curious to perhaps see it in your paper :)
> > > >
> > > >
> > > > While I maintain my concerns, I acknowledge that there could be some interesting ideas in the algorithm/proof (better placing in literature would help these stand out!) and I also appreciate your responses so I am willing to increase my score to 5.
> > > >
> > > > Thank you.

---

> > > > > ### Author Response · Authors · 2021-09-03
> > > > > **Thank you!**
> > > > >
> > > > > Yes, let's not get into an endless loop :-) While we respectfully disagree with the score, we greatly appreciate your detailed comments! We will incorporate these valuable comments in the revision, and we agree these changes will indeed further highlight our contributions and make the paper stronger. Again, our sincere thanks for the great suggestions!

---

### Decision · Program_Chairs · 2021-09-27

**Decision:**

Accept (Poster)

**Comment:**

This paper looks at linear stochastic bandits with unknown, non-linear constraints.

We had a discussion about the novelty of this paper, compared to the existing ones (with linear constraints). Even if the global idea is roughly the same, we believed that this paper still has new interesting methodology and that the non-linearity of constraints is actually quite challenging.

At the end, we believe this paper should pass the acceptance bar.